# Deep Bootstrap Aggregation via Least Squares Estimation

## Abstract

Bootstrap aggregation, commonly referred to as bagging, is a fundamental technique in ensemble learning designed to enhance the performance of predictive models. It is well-established that the effectiveness of bagging is strongly influenced by the management of correlations among the aggregated models. For instance, random forests, a widely-used ensemble method, address this issue by randomly selecting features to reduce the correlation between individual tree models. In this study, we propose a method called *Deep Bootstrap Aggregation* for regression tasks, which combines deep network architectures with least squares estimation to improve the predictive accuracy of bagging models. Both theoretical analysis and empirical experiments support the effectiveness of the proposed approach.

## 1 Introduction

Regression problems are frequently encountered in real-world scenarios, as they seek to uncover the complex relationships between a set of input variables and a continuous output variable. A conventional approach to addressing these challenges involves the use of statistical models, such as linear models or nonparametric regression techniques. Among the various nonparametric methods, *regression trees* are particularly popular for explaining the relationships between inputs and outputs.

Bishop (1995, chapter 9) and Breiman (1996) demonstrated the benefits of combining predictions from multiple models to enhance predictive performance. Bühlmann & Yu (2002) provided a theoretical analysis of bagging, introducing the concept of subbagging. Later, Breiman (2001) proposed the use of nearly uncorrelated models to further improve predictive accuracy. One notable example of this approach is the *Random Forests* algorithm. Introduced by Breiman (2001), random forests employ a strategy that aggregates predictions from multiple individual regression trees. This is accomplished through two primary techniques: bootstrapping the training data and randomly selecting subsets of input variables within each bootstrap. Specifically, each tree in the random forest is built using a bootstrapped dataset and a randomly chosen subset of input variables. The final prediction is computed as the (unweighted) average of the predictions from all individual regression trees. For a more in-depth understanding of random forests and their implementation, see Hastie et al. (2009), Biau (2012), and James et al. (2021).

In random forests, the random selection of input variables aims to reduce correlations among the bootstrapped regression trees, as lower correlations typically lead to improved predictive performance. However, it is important to note that random input variable selection does not always achieve this goal, making the unweighted average of predictions from individual regression trees suboptimal. Evidence supporting this is presented in Figure 1 in Section 2.

As emphasized by Ulaş et al. (2012), the development of a robust ensemble learning methodology hinges on four critical components: strategic selection of prediction models, precise hyperparameter tuning, careful use of data sampling techniques, and deliberate selection of input variables. Extensive efforts have been made to reduce correlations between prediction models, with a focus on increasing model diversity (Rosen, 1996; Ho, 1998; Liu & Yao, 1999; Derbeko et al., 2002; Kuncheva & Whitaker, 2003; Bacauskiene et al., 2009; Hsu & Srivastava, 2012) and refining ensemble weighting schemes (Acar & Rais-Rohani, 2009; Kim et al., 2011; Shahhosseini et al., 2020; Mao et al., 2021). Despite substantial research aimed at enhancing ensemble learning through these

components, as comprehensively reviewed by Tuysuzoglu & Birant (2020), there remains a significant gap in addressing the key issue of reducing correlations among prediction models.

To address this challenge, we begin by interpreting the unweighted sample mean in the bagging procedure as the ordinary least squares (OLS) estimator within a linear model that includes only an intercept term. To account for correlation among prediction models, it is advisable to replace OLS with generalized least squares (GLS) estimation, which leverages model correlations to improve efficiency. We propose a framework that incorporates least squares methodology within a deep network architecture. This framework effectively manages correlations across models by integrating hidden layers, leading to substantial improvements in predictive accuracy. Additionally, this approach extends beyond enhancing random forests, making it applicable to a wide range of models that handle continuous outputs. We demonstrate the robustness of the proposed method through theoretical analysis and empirical validation.

The structure of this article is as follows. Section 2 formalizes our aggregation method as a least squares estimation within the context of linear models. In Section 3, we introduce our framework and provide a detailed exposition of the corresponding theoretical results. The framework is extended to a deep network structure in Section 4. Sections 5 and 6 present the results of numerical studies, including both simulated and real-world examples, to validate the effectiveness of the proposed approach. Finally, conclusions are drawn in Section 7.

## 2 Aggregation as a Least Squares Estimation

Before introducing the proposed method, we first reformulate the bagging procedure by framing it as an estimation problem in linear models. Let the training dataset be $D = \{y_i, \mathbf{x}_i\}_{i=1}^n$, where $\mathbf{x}_i = (x_{i1}, x_{i2}, \ldots, x_{ip})^\top$ represents the vector of $p$ input variables for the $i$th data point, and $y_i = y(\mathbf{x}_i)$ is the observed response value corresponding to $\mathbf{x}_i$. The goal is to predict the output variable $y(\mathbf{x})$ at an unobserved $\mathbf{x}$, using the training dataset $D$.

Conventionally, the bagging procedure constructs $T$ predictors, $C_1, \ldots, C_T$, each trained on a bootstrapped dataset from $D$ under a chosen prediction model (e.g., regression tree, K-NN regression). We refer to $C_1, \ldots, C_T$ as the *base predictors* in this article. The aggregated prediction of $y(\mathbf{x})$ is given by:

$$\hat{y}(\mathbf{x}) = \frac{1}{T} \sum_{j=1}^{T} C_j(\mathbf{x}). \tag{1}$$

In Equation (1), each base predictor $C_j(\mathbf{x})$ is assigned an equal weight of $1/T$, meaning that all base predictors contribute equally to $\hat{y}(\mathbf{x})$. The use of Equation (1) is supported by Bishop (1995, Chapter 9) and Breiman (1996), who show that the mean square error (MSE) of $\hat{y}(\mathbf{x})$ is lower than that of any single predictor $C_j(\mathbf{x})$. This property makes bagging a widely adopted machine learning algorithm.

Although equally weighted aggregation, as described in Equation (1), is computationally efficient and easy to understand, it is important to critically reassess the assumptions underlying this approach. In the following discussion, we reformulate the aggregation step as a least squares problem, providing an alternative perspective on equally weighted aggregation.

### 2.1 Ordinary Least Squares-Based Aggregation

The goal of aggregation in ensemble learning is to make predictions by combining the base predictors $C_j(\mathbf{x})$, where $j = 1, 2, \ldots, T$. Let $\epsilon_j(\mathbf{x}) = C_j(\mathbf{x}) - \mu(\mathbf{x})$, where $\mu(\mathbf{x}) = \mathrm{E}(C_j(\mathbf{x}))$ for $j = 1, 2, \ldots, T$, resulting in the following expression:

$$C_j(\mathbf{x}) = \mu(\mathbf{x}) + \epsilon_j(\mathbf{x}), \quad j = 1, 2, \ldots, T, \tag{2}$$

where the expectation is taken with respect to the distribution of bootstrapped samples, and $\epsilon_j(\mathbf{x})$ represents a random error. For a given $\mathbf{x}$, the OLS estimator of $\mu(\mathbf{x})$ is:

$$\hat{\mu}_{\mathrm{OLS}}(\mathbf{x}) = \frac{1}{T} \sum_{j=1}^{T} C_j(\mathbf{x}),$$

which is equivalent to $\hat{y}(\mathbf{x})$ in Equation (1). Thus, the conventional bagging method described in Equation (1) can be interpreted as the OLS estimation of $\mu(\mathbf{x})$ under the linear model presented in Equation (2). We refer to this method as ordinary least squares-based aggregation (OLSA).

It is well known that the OLS estimator is the best linear unbiased estimator (BLUE) under the following two assumptions:

$$\text{Var}(\epsilon_j(\mathbf{x})) = \sigma^2 \quad \text{and} \quad \text{Cov}(\epsilon_i(\mathbf{x}), \epsilon_j(\mathbf{x})) = 0,$$
$$i, j = 1, 2, \ldots, T, \quad i \neq j.$$

However, the construction of $C_1(\mathbf{x}), \ldots, C_T(\mathbf{x})$ involves bootstrapping the data, which can lead to correlations between $C_i(\mathbf{x})$ and $C_j(\mathbf{x})$ when $i \neq j$. The most extreme case arises when two bootstrapped datasets are identical, resulting in the corresponding base predictors producing identical predictions.

To illustrate the covariance between base predictors, we employ a random-forest model with the random selection of $\lfloor \sqrt{p} \rfloor$ input variables. We estimate the variance-covariance matrix of the base predictors as follows:

$$\widehat{\text{Cov}}(C_i(\mathbf{x}), C_j(\mathbf{x})) = \frac{1}{n-1} \sum_{k=1}^{n} \left(C_i(\mathbf{x}_k) - y(\mathbf{x}_k)\right)\left(C_j(\mathbf{x}_k) - y(\mathbf{x}_k)\right). \tag{3}$$

We use the following three functions to generate the training and testing data:

1. `detpep108d`: This 8-dimensional function, originally presented by Dette & Pepelyshev (2010), exhibits substantial curvature in certain dimensions while others are scarcely utilized.

2. `grlee09`: Created by Gramacy & Lee (2009), this function involves six variables and exhibits significant oscillations as the input variables approach zero.

3. `friedman1`: One of the friedman functions introduced by Friedman (1991), `friedman1` consists of ten variables and exhibits considerable variation near zero.

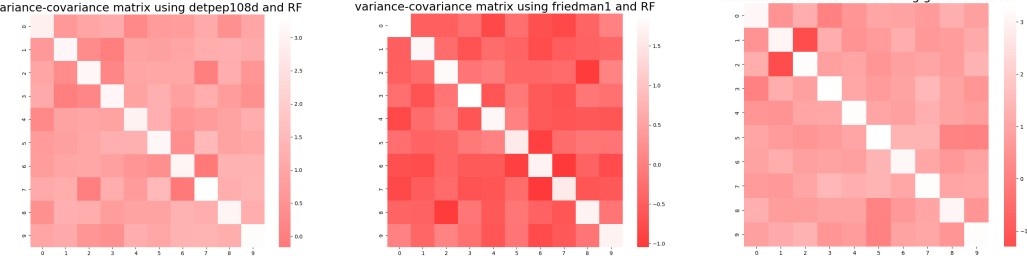

Figure 1: Comparison of the variance-covariance matrices for different underlying functions. The left, middle, and right panels correspond to the functions `detpep108d`, `friedman1`, and `grlee09`, respectively.

We generate 1000 training data points uniformly distributed in $[0, 1]$ for each input dimension and use the three functions, plus a random error term with zero mean and variance equal to 10 percent of the signal variance, to generate the output values. We construct $T = 10$ base predictors via 10 bootstrapped datasets. Then, we apply Equation (3) to estimate the variance and covariance between the 10 base predictors. The results are presented in Figure 1. To better visualize the variance-covariance matrices, we apply a log scale to each entry of the matrices. The off-diagonal elements represent the covariances between predictors, showing that predictors generated using all functions are correlated. Furthermore, predictors constructed from different underlying functions exhibit varying levels of correlation.

In the following section, we propose a new aggregation approach based on the linear-model framework in (2) to address the above issue. This method will be shown to significantly improve prediction performance via simulated and real-world data experiments.

## 3 GENERALIZED LEAST SQUARE-BASED AGGREGATION

Based on the observation in Figure 1, which displays varying magnitudes of variances and correlations among the base predictors, it is reasonable to improve the OLSA method using GLS estimation as follows:

$$\hat{\mu}_{\text{GLS}}(\mathbf{x}) = (\mathbf{1}_T^\top \mathbf{\Sigma}^{-1} \mathbf{1}_T)^{-1} \mathbf{1}_T^\top \mathbf{\Sigma}^{-1} \mathbf{C}(\mathbf{x}),$$

where $\mathbf{1}_T$ is the $T \times 1$ vector of ones, $\mathbf{C}(\mathbf{x})$ is the vector of $C_j(\mathbf{x})$ values for $j = 1, \ldots, T$, and $\mathbf{\Sigma}$ is the variance-covariance matrix of $\mathbf{C}(\mathbf{x})$. We refer to this method as Generalized Least Squares-based Aggregation (GLSA). It is known that $\hat{\mu}_{\text{GLS}}(\mathbf{x})$ is the BLUE of $\mu(\mathbf{x})$ under the model in Equation (2), according to the Gauss-Markov theorem, when $\mathbf{\Sigma}$ is not proportional to the identity matrix. This implies that the MSE of GLSA is lower than that of conventional bagging.

The matrix $\mathbf{\Sigma}$, estimated by Equation (3), accounts for the unequal variances and nonzero correlations between base predictors. It can be observed that:

$$\text{Var}(\hat{\mu}_{\text{GLS}}(\mathbf{x})) = \left(\mathbf{1}_T^\top \mathbf{\Sigma}^{-1} \mathbf{1}_T\right)^{-1} = \frac{1}{\sum_{i=1}^T \sum_{j=1}^T \mathbf{\Sigma}_{ij}^{-1}},$$

where $\mathbf{\Sigma}_{ij}^{-1}$ denotes the $ij$th element of $\mathbf{\Sigma}^{-1}$. Let $\mathbf{w}_{\text{GLS}} = (\mathbf{1}_T^\top \mathbf{\Sigma}^{-1} \mathbf{1}_T)^{-1} \mathbf{1}_T^\top \mathbf{\Sigma}^{-1}$ denote the vector of weights assigned to each base predictor in the computation of $\hat{\mu}_{\text{GLS}}(\mathbf{x})$. Notably, $\mathbf{w}_{\text{GLS}}$ depends solely on the variance-covariance matrix $\mathbf{\Sigma}$ constructed from the training data and is independent of the location of the testing data.

To examine the variance reduction achieved by employing GLSA compared to OLSA, we define $R_{\mathbf{\Sigma}}$ as a measure that quantifies the extent of variance reduction facilitated by GLSA:

$$R_{\mathbf{\Sigma}} = \frac{\text{Var}(\hat{\mu}_{\text{OLS}}(\mathbf{x})) - \text{Var}(\hat{\mu}_{\text{GLS}}(\mathbf{x}))}{\text{Var}(\hat{\mu}_{\text{OLS}}(\mathbf{x}))} = 1 - \frac{T^2}{\left(\sum_{i=1}^T \sum_{j=1}^T \mathbf{\Sigma}_{ij}^{-1}\right)\left(\sum_{i=1}^T \sum_{j=1}^T \mathbf{\Sigma}_{ij}\right)}.$$

Using the Cauchy-Schwarz inequality, it can be shown that:

$$\left(\sum_{i=1}^T \sum_{j=1}^T \mathbf{\Sigma}_{ij}^{-1}\right)\left(\sum_{i=1}^T \sum_{j=1}^T \mathbf{\Sigma}_{ij}\right) \geq T^4.$$

This demonstrates that $R_{\mathbf{\Sigma}}$ satisfies the inequality $R_{\mathbf{\Sigma}} \geq 1 - \frac{1}{T^2}$. The lower bound $1 - \frac{1}{T^2}$ increases with $T$, indicating that the variance reduction becomes more significant as $T$ grows. When equality holds, it implies that $\text{Var}(\hat{\mu}_{\text{OLS}}(\mathbf{x})) = T^2 \cdot \text{Var}(\hat{\mu}_{\text{GLS}}(\mathbf{x}))$, highlighting a substantial variance reduction through GLSA compared to OLSA.

In the remainder of this section, we examine the relationship between the number of predictors $T$ and the variances of $\hat{\mu}_{\text{OLS}}(\mathbf{x})$ and $\hat{\mu}_{\text{GLS}}(\mathbf{x})$. We use a random-forest model with $\lfloor \sqrt{p} \rfloor$ input variables. The training dataset, consisting of 400 observations, was generated using the functions `detpep108d`, `friedman1`, and `grlee09`. We begin with five base models and incrementally add five more in each subsequent iteration. For the GLSA method, the covariance matrix $\mathbf{\Sigma}$ is estimated using Equation (3).

Figure 2 presents the values of $R_{\mathbf{\Sigma}}$ and the variances of $\hat{\mu}_{\text{OLS}}(\mathbf{x})$ and $\hat{\mu}_{\text{GLS}}(\mathbf{x})$. To facilitate comparison, $R_{\mathbf{\Sigma}}$ and the variances of the predictors are plotted on the same graph. A key observation is that the proposed method significantly reduces the variance of the predictors, with the variance of $\hat{\mu}_{\text{GLS}}$ decreasing monotonically as $T$ increases. While the variances for both OLSA and GLSA decrease as $T$ increases in the `friedman1` and `detpep108d` functions, the rate of increase in $R_{\mathbf{\Sigma}}$ is slower, indicating that the improvement achieved by applying GLSA to these functions is less pronounced. In contrast, $R_{\mathbf{\Sigma}}$ increases consistently, suggesting that a larger number of predictors may lead to more substantial improvements using GLSA. In summary, increasing $T$ generally results in higher $R_{\mathbf{\Sigma}}$. Additionally, the trends of $R_{\mathbf{\Sigma}}$, $\text{Var}(\hat{\mu}_{\text{OLS}}(\mathbf{x}))$, and $\text{Var}(\hat{\mu}_{\text{GLS}}(\mathbf{x}))$ vary based on the choice of base models and the underlying functions.

## 4 DEEP BOOTSTRAP AGGREGATION VIA MULTI-LAYER GLSA

Numerous factors can influence the performance of GLSA. A critical issue is the potential instability encountered during the inversion of the variance-covariance matrix $\mathbf{\Sigma}$. This instability arises from

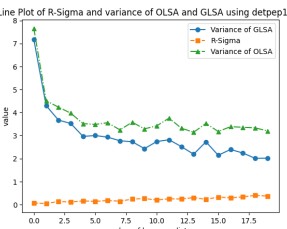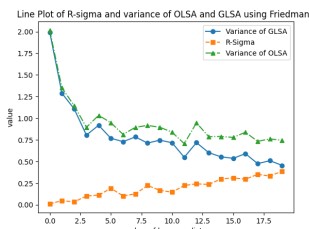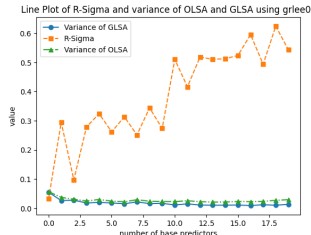

Figure 2: Comparison of $R_{\Sigma}$ and variance of predictors. The left, middle, and right panels employ the functions: `detpep108d`, `friedman1` and `grlee09` respectively.

numerical errors inherent in the inversion process, which can significantly impair the performance of GLSA. These numerical inaccuracies are especially problematic in high-dimensional settings or when $\Sigma$ is nearly singular, leading to unreliable estimates. In this section, we first demonstrate this difficulty and then propose a solution by incorporating a deep network structure into GLSA.

## 4.1 NON-STABLE PERFORMANCE FOR GLSA

Suppose we aim to aggregate the predictors $C_1, C_2, \ldots, C_T$. Figure 3 provides evidence of instability when using GLSA. We use the `friedman1`, `friedman2`, and `friedman3` functions with noise centered at zero and variance equal to 10 percent of the signal variance to sample 200 training data points, where `friedman2` and `friedman3` are also from Friedman (1991). Additionally, we use 400 testing data points to evaluate each model—bagging trees and random forests—and each aggregation method—OLSA and GLSA.

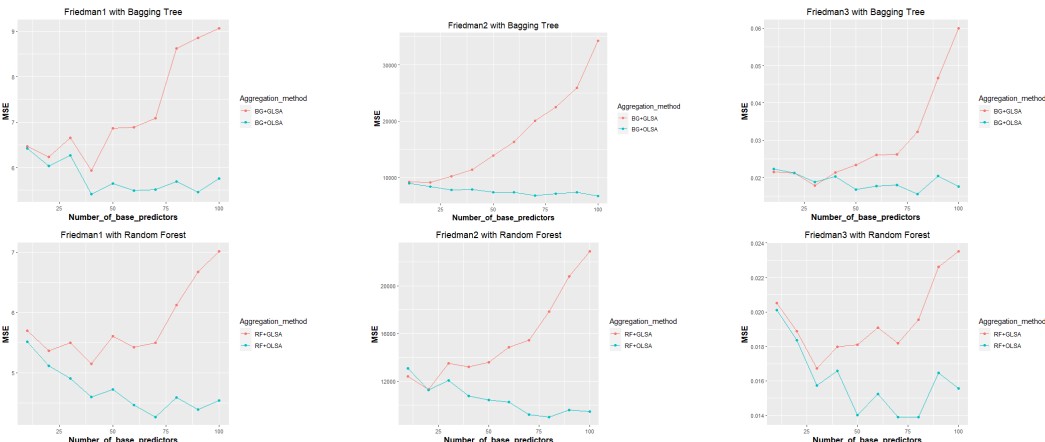

Figure 3: Trends in MSE performance as the number of predictors increases. The left, middle, and right panels correspond to `friedman1`, `friedman2`, and `friedman3`, respectively. The upper panels use bagging trees, while the lower panels use random forests.

For OLSA, it is clear that the MSE of bagging trees and random forests decreases as the number of predictors increases, consistent with the intuitive understanding that aggregating more predictors improves overall prediction accuracy. For GLSA, however, in certain cases—such as `friedman3` with random forests—the MSE reaches a minimum at specific points and then increases as the number of predictors continues to grow. Even more concerning, in cases like `friedman2` with bagging trees, the MSE shows a monotonically increasing trend, which is undesirable.

To address this issue, we propose an enhanced modification of GLSA, called Multi-Stage Generalized Least Squares-Based Aggregation (M-GLSA), by incorporating a deep network structure into the aggregation procedure. This approach is able to effectively combine base predictors from different stages of the aggregation process, improving stability and further enhancing predictive accuracy.

## 4.2 MULTI-STAGE GLSA

A deep network consists of several layers. In this article, we treat each layer as a stage in the aggregation procedure. M-GLSA aggregates a subset of predictors using GLSA at each stage and produces a final predictor after the last stage. Let $\mathbf{C} = (C_1, \ldots, C_T)$ represent the set of base predictors to be aggregated. At stage 1, we divide $\mathbf{C}$ into $T_1$ subsets, where each subset contains $T_{1,1}, \ldots, T_{1,T_1}$ base predictors. We use $T_{i,j}$ to denote the number of predictors in the $j$th subset at the $i$th stage. At stage 1, we aggregate the predictors in each subset using GLSA to obtain the stage 1 predictors $\mathbf{C}_1 = (C_{1,1}, C_{1,2}, \ldots, C_{1,T_1})$, where $C_{1,j}$ denotes the $j$th predictor at stage 1. Similarly, we divide $\mathbf{C}_1$ into $T_2$ subsets, with each subset containing $T_{2,1}, \ldots, T_{2,T_2}$ predictors. GLSA is then applied to obtain the stage 2 predictors $\mathbf{C}_2 = (C_{2,1}, \ldots, C_{2,T_2})$. Recursively, this process continues, with GLSA being performed at each stage using appropriately sized subsets, until the final predictors are obtained.

Note that each stage consists of multiple aggregation steps, with each step involving only a subset of the predictors from the previous stage. This property reduces the dimensionality of the variance-covariance matrix $\mathbf{\Sigma}$ in each step and provides a more accurate estimate of $\mathbf{\Sigma}$. However, M-GLSA has several hyperparameters, including the number of predictors $T$, the number of stages $G$, the subset sizes at each stage $T_{1,1}, \ldots, T_{i,T_i}$ for $i = 1, \ldots, G$, and the number of subsets at each stage $T_1, \ldots, T_G$. In the following, we discuss the selection of suitable hyperparameters, taking into account computational time, performance, and numerical stability.

Recall that when selecting the number of base predictors $T$ in bagging trees and random forests, the goal is to choose a value of $T$ large enough to reach a stable MSE. We adopt the same approach in M-GLSA. After completing the $k$th stage, the predictors $\mathbf{C}_k$ are more accurate compared to those in $\mathbf{C}_i$, $i = 1, \ldots, k-1$. This is evident because the number of base predictors used to construct each predictor at stage $k$ is greater than that in the preceding stages. However, if the number of stages $G$ is chosen to be large, the predictors in the final stages may potentially lead to a singular variance-covariance matrix. In our experience, two stages ($G = 2$) are sufficient to reduce the subset size at each step to approximately $\sqrt{T}$, assuming a suitable choice of $T$. Based on this observation, we focus on two-stage GLSA, referred to as 2-GLSA. Figure 4 illustrates the comparison between GLSA and 2-GLSA.

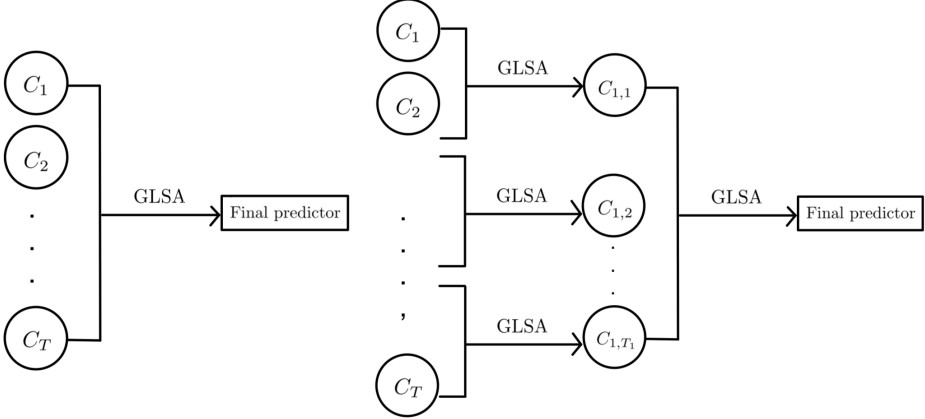

Figure 4: Comparison of the two GLSA methods. The left-hand side shows the diagram of GLSA, while the right-hand side illustrates the diagram of 2-GLSA.

From the above discussion, we suggest that 2-GLSA with a sufficiently large number of predictors $T$ is sufficient for producing stable predictors. For selecting the subset size at each stage, let us assume the variance-covariance matrix $\mathbf{\Sigma} = \sigma^2 \mathbf{I}_T$ and consider OLSA. Suppose we apply two-stage OLSA as with 2-GLSA to aggregate $T$ predictors $\mathbf{C} = (C_1, \ldots, C_T)$. Let $T_{1,1}, \ldots, T_{1,T_1}$ denote the sizes

of the subsets at stage 1, where $\sum_{j=1}^{T_1} T_{1,j} = T$. The predictors after stage 1 become

$$C_{1,t} = \sum_{j=1}^{T_{1,t}} \frac{1}{T_{1,t}} C_{\left(\sum_{k=1}^{t-1} T_{1,k}\right)+j}, \quad \forall t \in 1, \ldots, T_1.$$

Note that the predictor $C_{1,t}$ at stage 1 is simply the average of the predictors in the $t$th subset. The predictors $C_{1,t}, \ t = 1, \ldots, T_1$, still have the same variance scale and remain uncorrelated. Hence, we can derive the final predictor as follows:

$$\sum_{t=1}^{T_1} \frac{1}{T_1} C_{1,t} = \sum_{t=1}^{T_1} \sum_{j=1}^{T_{1,t}} \frac{1}{T_1 T_{1,t}} C_{\left(\sum_{k=1}^{t-1} T_{1,k}\right)+j}.$$

Given the assumption $\boldsymbol{\Sigma} = \sigma^2 \mathbf{I}_T$, if we want to achieve the lowest variance of the predictor produced by 2-OLSA, the subset sizes in stage 1 should be as equal as possible. Therefore, we recommend that the subset sizes in stage 1 be identical. Specifically, $T_{1,j} = T/T_1$ for all $j = 1, \ldots, T_1$. In this case, there exists an integer $S$ such that $S = T/T_1$, and we only need to determine $S$. A heuristic method for choosing $S$ and $T_1$ is to find a pair of values such that both are close to $\sqrt{T}$. This aims to balance the number of predictors allocated to each stage. If $S$ or $T_1$ is chosen far from $\sqrt{T}$, stability may not be achieved in either stage 1 or stage 2, as the number of predictors allocated to one stage could become excessively large. The most extreme case occurs when $S$ is chosen as 1 or $T$, resulting in instability in stage 2 and stage 1, respectively. We use Algorithm 1 below to illustrate the 2-GLSA procedure.

---

**Algorithm 1** 2-GLSA

---

**Require:** Base predictors $C_1 \ldots, C_T$, training set $D$, number of subsets $S$
1: $T_1 = T/S$                        ▷ Number of subsets in stage 1
2: Randomly split the base predictors into $S$ subset with equal size.
3: **for** $t = 1$ to $T_1$ **do**                      ▷ For each subset
4:      Use the base predictors in subset $t$ to estimate the variance-covariance matrix, denoted by $\boldsymbol{\Sigma}_t$, via Equation (3).
5:      $\mathbf{W}_t = (\mathbf{1}_S^\top \boldsymbol{\Sigma}_t^{-1} \mathbf{1}_S)^{-1} \mathbf{1}_S^\top \boldsymbol{\Sigma}_t^{-1}$
6:      Obtain $C_{1,t}$ by weighted averaging the base predictors in subset $t$ using $\mathbf{W}_t$
7: **end for**
8: Use $C_{1,t}, \ t = 1, ..., T_1$ to estimate the variance-covariance matrix, denoted by $\boldsymbol{\Sigma}$, via Equation (3).
9: **Return** $(\mathbf{1}_{T_1}^\top \boldsymbol{\Sigma}^{-1} \mathbf{1}_{T_1})^{-1} \mathbf{1}_{T_1}^\top \boldsymbol{\Sigma}^{-1} (C_{1,1}, ..., C_{1,T_1})^\top$

---

## 5    SIMULATION STUDIES

In this section, we conduct a comparative analysis using simulation data to assess the performance of 2-GLSA in comparison with random forests. We conduct experiments using six simulated datasets obtained from the Virtual Library of Simulation Experiments (VLSE; Surjanovic & Bingham (2023)). Our primary objective is to evaluate the performance of random forests under various ensemble procedures and different numbers of base predictors. Specifically, we utiliz the `friedman1`, `friedman2`, `friedman3`, `grlee09`, and `detpep108d` functions, as discussed in earlier sections. Additionally, we include the `welchetal92` function, a simulation model with 20 input variables introduced by Welch et al. (1992). Table 1 summarizes these functions along with the number of training and testing data points.

To evaluate the efficacy of the proposed methodology, we conduct 100 randomized experiments, each involving the generation of new training and testing datasets. In these experiments, random noise with zero mean and a variance equal to 10% of the signal variance is added to both the training and testing data, drawn from a uniform distribution on $(0, 1)$. We use OLSA as the benchmark for comparison with 2-GLSA. Three choices of the number of base predictors $T$ are considered:

Table 1: Summary of simulation data

| Function | Dimension | Number of training data | Number of testing data |
|---|---|---|---|
| welchetal92 | 20 | 800 | 2000 |
| detpep108d | 8 | 200 | 2000 |
| grlee09 | 6 | 200 | 2000 |
| friedman1 | 10 | 200 | 2000 |
| friedman2 | 4 | 200 | 2000 |
| friedman3 | 4 | 200 | 2000 |

$20, 50, 100$. Additionally, we also assess the performance of M-GLSA, where a complete factorization of the number of base predictors is used to determine the subset size at each stage of aggregation. For example, with $T = 20$ base predictors, the aggregation process is organized into three stages due to $20 = 2 \times 2 \times 5$:

- **Stage 1:** Produce 10 aggregated predictors, each derived from the aggregation of 2 base predictors.

- **Stage 2:** From the 10 predictors generated in Stage 1, produce 5 aggregated predictors, each resulting from the aggregation of 2 predictors from Stage 1.

- **Stage 3:** In the final stage, all 5 predictors from Stage 2 are aggregated to form the final predictor.

The above procedure is also applied to $T = 50 = 2 \times 5 \times 5$ and $T = 100 = 2 \times 2 \times 5 \times 5$ for M-GLSA.

Figure 5 provides a comprehensive summary of performance across six different functions, depicted through boxplots. The performance metrics are presented on a logarithmic scale of MSE. Table 2 summarizes the MSE and its confidence interval across all functions and methods. The results demonstrate significant improvements with the 2-GLSA method, particularly for the functions welchetal92 and detpep108d. Notably, 2-GLSA consistently outperforms OLSA across nearly all functions and settings. The trend indicates that 2-GLSA reduces MSE as the number of base predictors increases, similar to the behavior observed with OLSA. Additionally, while M-GLSA shows superior performance over 2-GLSA in certain scenarios, it is constrained by its higher computational time due to the multiple executions of GLSA.

Table 2: Summary of performance with simulation data using confidence of MSE

| Functions | # of trees | Methods | | |
|---|---|---|---|---|
| | | M-GLSA | 2-GLSA | OLSA |
| welchetal92 | 20 | $\mathbf{27.6208 \pm 0.2786}$ | $27.6457 \pm 0.3044$ | $29.3192 \pm 0.3559$ |
| | 50 | $26.2047 \pm 0.2573$ | $\mathbf{26.1688 \pm 0.2648}$ | $27.6145 \pm 0.3114$ |
| | 100 | $25.7058 \pm 0.2565$ | $\mathbf{25.5648 \pm 0.2451}$ | $27.0280 \pm 0.2853$ |
| detpep108d | 20 | $\mathbf{39.9922 \pm 0.5177}$ | $39.8543 \pm 0.5325$ | $40.2140 \pm 0.5782$ |
| | 50 | $\mathbf{37.9753 \pm 0.4936}$ | $37.9961 \pm 0.5043$ | $38.1998 \pm 0.5036$ |
| | 100 | $\mathbf{37.3278 \pm 0.5101}$ | $37.3399 \pm 0.5136$ | $37.5388 \pm 0.5184$ |
| grlee09 | 20 | $\mathbf{0.2611 \pm 0.0045}$ | $0.2628 \pm 0.0047$ | $0.2772 \pm 0.0052$ |
| | 50 | $0.24699 \pm 0.0046$ | $\mathbf{0.2450 \pm 0.0041}$ | $0.2623 \pm 0.0042$ |
| | 100 | $0.2426 \pm 0.0039$ | $\mathbf{0.2351 \pm 0.0041}$ | $0.2599 \pm 0.0039$ |
| friedman1 | 20 | $11.8795 \pm 0.1512$ | $\mathbf{11.7927 \pm 0.1519}$ | $11.8502 \pm 0.1457$ |
| | 50 | $\mathbf{11.2672 \pm 0.1354}$ | $11.2584 \pm 0.1431$ | $11.3068 \pm 0.1409$ |
| | 100 | $11.0430 \pm 0.1268$ | $\mathbf{11.0251 \pm 0.1305}$ | $11.1576 \pm 0.1392$ |
| friedman2 | 20 | $41520 \pm 632$ | $\mathbf{41436 \pm 625}$ | $41598 \pm 651$ |
| | 50 | $\mathbf{39626 \pm 617}$ | $39657 \pm 592$ | $39734 \pm 600$ |
| | 100 | $\mathbf{38989 \pm 590}$ | $39313 \pm 589$ | $39263 \pm 578$ |
| friedman3 | 20 | $0.04387 \pm 0.00099$ | $0.04389 \pm 0.00113$ | $\mathbf{0.04364 \pm 0.00103}$ |
| | 50 | $0.04235 \pm 0.00101$ | $\mathbf{0.04189 \pm 0.00094}$ | $0.04213 \pm 0.00097$ |
| | 100 | $0.04160 \pm 0.00092$ | $0.04193 \pm 0.00102$ | $\mathbf{0.04152 \pm 0.00095}$ |

## 6 APPLICATIONS

We demonstrate two applications of the proposed 2-GLSA. The first is a real-world data application, and the second addresses a subsampling issue in big data.

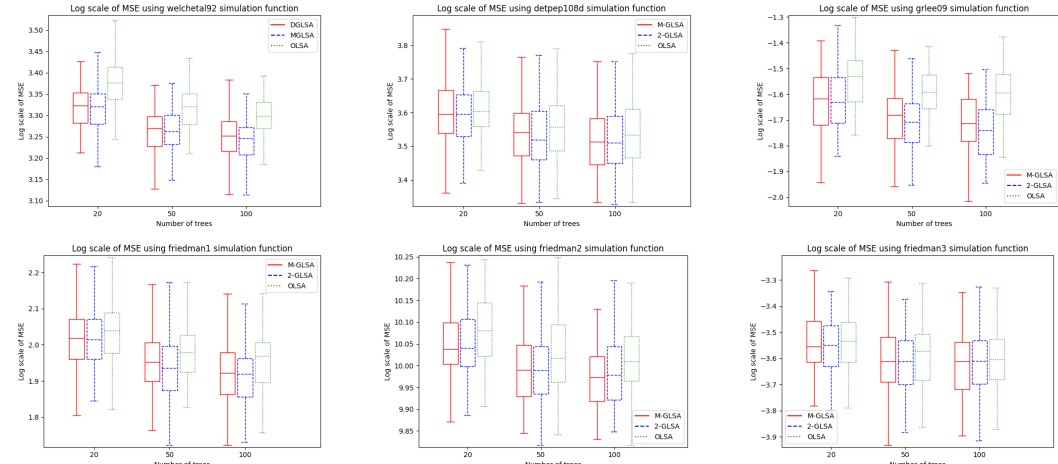

Figure 5: Comparison of prediction performance on simulation data. The figure shows boxplots of the MSE on a logarithmic scale.

## 6.1 REAL DATA STUDIES

We utilize two datasets, namely the Boston Housing dataset and the Concrete Compressive Strength dataset, to illustrate the practical application of 2-GLSA. The Boston Housing dataset, originally documented by Harrison & Rubinfeld (1991), is a widely adopted benchmark in machine learning studies and comprises 506 samples characterized by 12 features. The Concrete Compressive Strength dataset, introduced by Yeh (2007), contains 1030 samples with 8 features. This dataset is particularly suited for exploring ensemble learning techniques due to the intricate and interconnected nature of its features.

We conduct 100 random divisions of training and testing datasets, with 20% of the samples used as the testing set. For this analysis, we utilize 1000 base predictors ($T = 1000$). As in Section 5, we consider three aggregation mehods: OLSA, 2-GLSA, and M-GLSA with omplete factorization. Table 3 presents the MSE and its confidence interval across the two datasets and three aggregation methods. The 2-GLSA method consistently outperforms other aggregation techniques. Additionally, the confidence intervals associated with the MSE for 2-GLSA are notably smaller compared to the other methods in both datasets.

Table 3: Summary of performance (MSE) in real data

| Dataset | M-GLSA | 2-GLSA | OLSA |
|---|---|---|---|
| Boston | $10.52 \pm 1.52$ | $\mathbf{10.27 \pm 1.39}$ | $10.70 \pm 1.51$ |
| concrete | $30.31 \pm 2.27$ | $\mathbf{26.82 \pm 2.14}$ | $33.98 \pm 2.47$ |

## 6.2 SUBSAMPLING

Numerous studies have explored subsampling methods designed to select a subset of data points from large datasets. For instance, Joseph & Mak (2021) introduced a robust supervised subsampling method referred to as *supercompress* that demonstrates resilience across various statistical models. In this subsection, we apply the 2-GLSA method to aggregate base predictors constructed from a subset of the full dataset, with OLSA and supercompress used as benchmarks. We use 1-Nearest-Neighbor as the base model. Our focus is on predicting the seventh output of the SARCOS dataset, as previously examined in Rasmussen & Williams (2006), utilizing all 21-dimensional inputs. In the experiments, a random subset of 500, 1000, 1500, and 2000 out of 44,484 data points is sampled for training in each iteration, with the exception of supercompress, which is itself a well-established subdata selection technique. We generate $T = 10$ and $T = 20$ base predictors, respectively. A pre-specified testing dataset of 449 data points, suggested by Rasmussen & Williams (2006), is used to evaluate the performance of each method.

Figure 6 illustrates the MSE results. The results indicate that OLSA performs significantly better than supercompress, regardless of the number of base models aggregated. Notably, 2-GLSA outperforms OLSA. This suggests that the proposed 2-GLSA is a promising approach and can be effectively adopted in a subsampling framework within the context of big data.

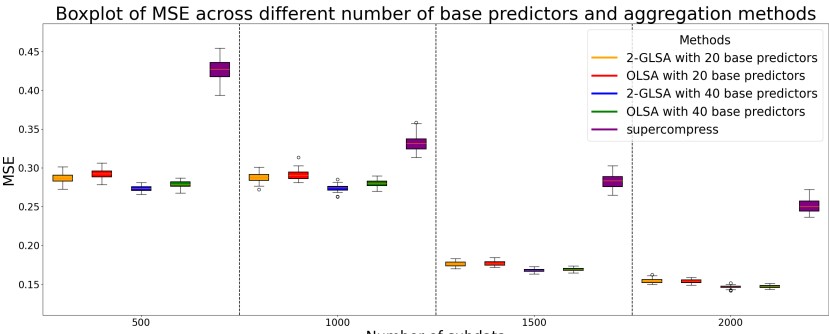

Figure 6: Boxplot of MSE across different numbers of base predictors and aggregation methods. The boxplot displays the MSE for five different methods (2-GLSA with 20 base predictors, OLSA with 20 base predictors, 2-GLSA with 40 base predictors, OLSA with 40 base predictors, and supercompress) across four subsampling sizes (500, 1000, 1500, and 2000).

## 7 CONCLUSION AND FUTURE WORK

Previous research primarily focused on mitigating predictor correlations through algorithmic modifications or the development of sampling strategies. In contrast, the primary innovation of this study lies in directly addressing predictor correlations within the context of ensemble learning. Our approach leverages the training data not only to build predictors but also to extract valuable information for estimating predictor correlations. Empirical evidence from both theoretical analysis and numerical experiments confirms that our method significantly reduces MSE in prediction. One particularly noteworthy outcome is the substantial improvement observed in the random-forest models when applying Generalized Least Squares-based Aggregation, paving the way for new applications of random forests. Importantly, our methodology is simple to implement and does not require parameter tuning. Additionally, we introduce a computationally efficient statistic that facilitates the assessment of expected improvements when using Generalized Least Squares-based Aggregation.

Correlations manifest across various aspects of ensemble learning, and these phenomena extend to other applications as well. For example, in cross-validation, the average loss is computed from models constructed using overlapping subsets of training data. Similarly, in stochastic gradient descent, random sampling of subsets of observed data determines the direction sequentially. When the selected subsets overlap, the directions tend to align, leading to high correlations. A potential avenue for future research involves addressing correlations in other application domains beyond ensemble learning. It is also worth noting that the correlation patterns observed in this study vary depending on the datasets and base predictors used. Therefore, another promising direction for future research is to explore the origins of these correlations and deepen our understanding of the underlying mechanisms.

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
