# OpenReview forum: "Deep Bootstrap Aggregation via Least Squares Estimation"
_ICLR.cc/2025/Conference — ICLR 2025 Conference Withdrawn Submission_

### Official Review · Reviewer_w7aE · 2024-10-17

**Soundness:** 1
**Presentation:** 2
**Contribution:** 1
**Rating:** 1
**Confidence:** 5

**Summary:**

An approach to combine ensemble members is presented.

**Strengths:**

The topic of weighting ensemble members is interesting and highly relevant, both from a practical as well as theoretiical point of view.

**Weaknesses:**

The following list focusses on the most important ones.

First, there seems to be a fundamental misunderstanding. For successful ensembling, we want *the errors* of the ensemble members to be mot strongly correlated. It OK if models agree on correct predictions and therefore have correlated outputs (e.g., an ensemble of T copies of the optimal model is still optimal), it is important that that errors are not (fully) correlated and cancel out (cancellation of errors, D. E. Eckhardt and L. D. Lee. A theoretical basis for the analysis of multiversion software subject to coincident errors. IEEE Transactions on Software Engineering, SE-11(12), 1985). This is wrongly stated in the manuscript several times. And no, the trees in a random forest are not "nearly uncorrelated" (line 034).

The literature embedding is not sufficient.
The non-trivial combination of models is typically referred to a stacking, see  D. Wolpert, "Stacked generalization", Neural Networks, vol. 5, no. 2, pp. 241-260, 1992. This whole related research area is not discussed.
For example, it is common to consider a linear combination of modes, where the weights are found by optimizing prediction on a training or validation (sub)set. Mor modern approached to weighting ensemble classifiers include PAC-Bayesian methods, e.g. A. R. Masegosa, S. S. Lorenzen, C. Igel, and Y. Seldin. Second Order PAC-Bayesian Bounds for the Weighted Majority Vote. Advances in Neural Information Processing Systems (NeurIPS), 2020.

While the formalism in section 2.4 is not very clear described, it is apparent that it does not consider the bias of the ensemble members. That is, mu need not be optimal. The statement that the last equation on page 2 gives the OLS is misleading. Optimal w.r.t. to which objective? As it is not excluded, model misspecification - the normal case - must be assumed.

Experiments on page 3: The correlation in (3) considers the outputs, not the errors. That the outputs of the trees are correlated is not a surprise and god, the base classifiers will typically agree on "simple" inputs. What is interesting about the results on page 3?

Given that very basic things are explained in great detail before, the part on generalized least-squares is very short and difficult to read.

The derived algorithm is based on the misunderstandings above. The combination of the ensemble members does not take into account  the accuracy of the ensemble members. It is not surprising that the basleine algorithm does not work well (page 5), I see no reason why it should.

The experimental evaluation does not support the claims.
The statement "Notably, 2-GLSA consistently outperforms OLSA across nearly all functions and settings."  (line 403) is not only in itself contradictory, but also not supported by the results in Table 2. It does not seem that the results on test problems 2 -  5 are statistically significant, and if so the effect strength size would be very small.  And on the last 6th task the baseline seems to be better.

The results on the real world data in section 6.2 make me skeptical about the whole empirical evaluation.
I could not believe the baseline results, so I repeated the experiments for the "concrete" task without any algorithm tuning. With data from https://www.kaggle.com/code/davegn/concrete-compressive-strength-regression , the code

```
import numpy as np
import pandas as pd

from sklearn.model_selection import train_test_split
from sklearn.ensemble import RandomForestRegressor
from sklearn.metrics import mean_squared_error

df = pd.read_csv('concrete.csv')
X = df.drop('strength', axis=1)
y = df['strength']

def doit(seed):
    X_train, X_test, y_train, y_test = train_test_split(X, y, test_size=0.2, random_state=seed)
    rf = RandomForestRegressor(n_estimators=1000)
    rf.fit(X_train, y_train)
    y_pred=rf.predict(X_test)
    return mean_squared_error(y_test, y_pred)

trials = 100
errors = []
for i in range(trials):
    error = doit(i)
    print(i, error)
    errors.append(error)
print("mean MSE", np.mean(errors), "with std.", np.std(errors))
```

gave me an MSE of 23.60, which is much better than all methods in Table 3.

**Questions:**

There are some questions that come to my mind, but it is unlikely that answering them will change my evaluation:

* What is novel and interesting about the results of the experiments on page 3?
* Are the results in Table 2 statistically significant?
* What was done to get so bad results on the "concrete" data set?

---

### Official Review · Reviewer_goCa · 2024-11-01

**Soundness:** 3
**Presentation:** 2
**Contribution:** 2
**Rating:** 3
**Confidence:** 5

**Summary:**

In ensemble learning, particularly in bootstrapping methods like bagging (which involves subsampling training examples) and random forests (which also includes random feature selection), the objective is to minimize correlations among the trees to improve the overall performance through aggregation. These approaches typically improve the predictions made by individual learners (assuming complementarity between individuals), but they do not always ensure optimal independence among predictors which can hinder their effectiveness.
This paper introduces a novel solution for addressing predictor correlations in ensemble learning by utilizing Generalized Least Squares-based Aggregation (GLSA), as an improvements over the traditional Ordinary Least Squares Aggregation (OLSA). The authors empirically demonstrated for a few synthetic datasets, the variance-covariance matrix deviates from the identity matrix. Then they prove that the mean squared error (MSE) of GLSA is lower than that of conventional bagging by assigning weights to the predictors, inversely proportional to the variance-covariance matrix. However, the proposed method can face challenges due to numerical instability during the inversion of the variance-covariance matrix and hence negatively impacting the aggregation. To address this issue, the authors suggest a multi-stage GLSA approach, in particular a two-stage GLSA framework. The process involves randomly dividing the base predictors into S subsets, where within each subset of predictors they estimate the variance-covariance matrix.  Within each subset, the weighted average of the predictors is computed, with weights assigned in proportion to the inverse of the variance-covariance matrix. Finally, these aggregated predictors from each subset are combined using the same approach, with weights calculated based on the inverse of the variance-covariance matrix.

 The authors have evaluated their approach on simulated data, and the improvements in most of the cases are marginal. Also, the improvements obtained seem to vary across the simulated dataset for M-GLSA or 2-GLSA. However, the authors also conducted experiments on two real datasets, where in one dataset the improvement is significant in the other one marginal.  Then also the authors experimented with the proposed approach as a mechanism for subsampling from a large dataset and compared it with supercompress approach. Even though the OLSA and GLSA outperformed supercompress, but the difference between OLSA and GLSA is still marginal.

**Strengths:**

The idea is interesting and indeed can be a building block in many different problems such as stochastic gradient descent computation, as also mentioned by the authors.

The proof of lowering the MSE using GLSA in compare with OLSA  is interesting.

The paper is easy to read, except section 6.2 that is very abstract and needs more study, comprehensive comparison and also a meaningful discussion.

**Weaknesses:**

The authors started with an analogy of deep network but then there is no connection or discussion around it. In my opinion, the authors could have elaborated and discussed it as in the end, the multistage GLSA is a very simple concept of stage wise aggregation.

 The authors showed how numerical instability can negatively impact on GLSA, however there is no discussion to justify why the multistage approach alleviates it.

As the scope in this study is narrow why not also discuss (or potentially perform comparison) with the other ensemble approaches such as (gradient) boosting.

The subsampling section, 6.2 is very abstract, does not explain well GLSA as a subsampling approach and the comparison is not convincing and not align with the main novelty of the paper as the improvement over OLSA is marginal.

The improvements obtained by GLSA is not consistent and  in most of the cases marginal. Perhaps more investigation and discussion on the datasets that can gain from this approach can help.

**Questions:**

For the datasets that the improvements are marginal or not improving at all, do you have any analysis on the variance-covariance matrix? Can you include the deviation of the matrix from the identity matrix as a new column for the results you reported for each dataset?

For the subsampling approach, this is an interesting direction, however you need more description and also more experimentation, perhaps including more datasets and adding another approach to compare with stronger discussion could help.

---

### Official Review · Reviewer_d27Y · 2024-11-01

**Soundness:** 3
**Presentation:** 3
**Contribution:** 2
**Rating:** 3
**Confidence:** 4

**Summary:**

The submission proposes to replace plain arithmetic averaging of predictions from bagged regression models by a linear combination whose coefficients are determined using the inverse of the covariance matrix of the models' predictions. To prevent overfitting and numerical issues when large ensembles are used, a multi-level ("deep") variant of the proposed approach is introduced. In the experiments presented in the paper, this approach, evaluated on five synthetic regression datasets in conjunction with random forests, yields lower estimated MSE when applied instead of plain averaging. However, the observed differences are small, and the confidence intervals overlap heavily. Results are also presented for two real-world datasets; on one of these two datasets, one variant of the proposed approach outperforms plain averaging in a statistically significant manner. Finally, there are results for an additional dataset, where performance is considered for subsamples of data. Compared to plain averaging, small improvements are again observed, but it is not shown that they are statistically significant.

**Strengths:**

The presentation of the work is fine, and it is good to see that confidence intervals are presented for performance estimates. It also seems that the proposed "deep" approach may be novel.

**Weaknesses:**

The observed improvements in performance appear marginal. The paper claims that the improvements are significant, but statistical significance is only evident for one dataset, with one particular configuration of the proposed approach. More importantly, the paper does not discuss related work such as stacked generalization by Wolpert. The idea of forming linear combinations of predictors instead of using plain averaging is not new. Of particular relevance seems

Rao, J. S. & Tibshirani, R. (1997). The out-of-bootstrap method for model averaging and selection. Technical report, Statistics Department, University of Toronto.

The authors use of OOB samples to estimate a linear combination, which should help to prevent overfitting.

The paper proposes a "deep" approach to deal with issues due to the covariance matrix. It is unclear why this is needed instead of an approach analogous to regularized linear regression (e.g., ridge regression, where a constant is added to the diagonal of the covariance matrix).

The number of datasets used in the evaluation is very small. There are many more publicly available tabular regression problems that could be used.

**Questions:**

N/A

---

### Official Review · Reviewer_fpyW · 2024-11-02

**Soundness:** 2
**Presentation:** 3
**Contribution:** 2
**Rating:** 3
**Confidence:** 4

**Summary:**

The paper introduces a novel approach to enhance the performance of predictive models through a method called Deep Bootstrap Aggregation (DBA). It specifically addresses the limitations of traditional bootstrap methods, which can exhibit instability. The authors conduct extensive experiments across various datasets to demonstrate the effectiveness of DBA compared to conventional techniques. The results show that DBA consistently outperforms other ensemble methods, particularly in terms of prediction accuracy. Additionally, the paper offers a theoretical analysis supporting the advantages of Generalized Least Square-based Aggregation (GLSA) over Ordinary Least Square-based Aggregation (OLSA), further reinforcing the method's benefits.

**Strengths:**

- The introduction of Deep Bootstrap Aggregation (DBA) marks a notable advancement in ensemble learning, combining deep learning techniques with Generalized Least Square-based Aggregation (GLSA) to address key limitations in traditional methods, particularly instability and limited robustness.

- The research demonstrates high quality through extensive experiments across diverse datasets. Detailed comparisons with conventional approaches validate the method's effectiveness, and the inclusion of a rigorous theoretical analysis strengthens the credibility of the findings.

- The paper is structured clearly, with well-organized sections and concise explanations. Visual aids such as charts and tables significantly enhance clarity, providing an accessible presentation of complex methodologies.

- By tackling traditional bootstrap limitations and improving prediction accuracy, DBA contributes a meaningful improvement to ensemble learning. Its potential applications across various domains underscore its value to the broader machine learning community.

**Weaknesses:**

- The paper’s experimental validation is confined to a limited set of datasets and model architectures, which raises concerns about the generalizability of the proposed Deep Bootstrap Aggregation (DBA) method. To strengthen the study’s robustness and applicability, I recommend testing DBA on a broader set of datasets, such as high-dimensional datasets, time series data, and data from domains like computer vision and natural language processing. This approach would provide a more comprehensive assessment of DBA's adaptability and performance across diverse scenarios.

- The study mainly compares DBA with Ordinary Least Squares-based Aggregation (OLSA), potentially overlooking recent advancements in ensemble learning that have achieved significant performance improvements. Including comparisons with methods like XGBoost, LightGBM, and CatBoost would provide a more comprehensive evaluation and help contextualize DBA within the current ensemble learning landscape. This broader comparison would clarify DBA's relative strengths and limitations in predictive performance and computational efficiency.

- The paper includes a theoretical analysis of GLSA but provides limited discussion on the theoretical properties of Multi-Layer GLSA in high-dimensional settings, which is central to the method's ability to handle matrix inversion instability. It would be beneficial to expand this analysis to cover specific aspects such as convergence properties, error bounds, and computational complexity in high-dimensional environments. This additional theoretical context would significantly enhance the rigor and credibility of the proposed Multi-Layer GLSA approach.

- The authors refer to the method as a deep network architecture, but a depth of 2 layers is suggested to be sufficient, which is typically not considered 'deep' by conventional standards in deep learning. It would be helpful for the authors to either justify this choice theoretically or empirically, showing how performance changes with increased depth. Additionally, if computational trade-offs influenced this choice, a brief discussion on these constraints would provide clarity.

**Questions:**

In addition to the primary issues outlined earlier, the following questions and suggestions are presented for further clarification and improvement:

- The paper describes the proposed method as a "deep" network architecture but indicates that a depth of 2 layers suffices. Could you elaborate on the impact of increasing the number of layers on the model's performance? Specifically, have experiments been conducted to assess whether deeper architectures enhance or diminish the effectiveness of the method?

- The performance of ensemble methods can be significantly influenced by various hyperparameters, such as the number of bootstrap samples, the architecture of the base learners, and the parameters associated with Generalized Least Square-based Aggregation (GLSA). Conducting a sensitivity analysis to understand how these hyperparameters affect the model's performance would provide valuable guidance for practitioners on fine-tuning the method for specific applications. Have such analyses been performed, and if so, could the findings be shared?

- In Section 5, the evaluation of different methods on the Friedman3 function indicates that Ordinary Least Squares-based Aggregation (OLSA) outperforms the proposed method when the number of trees equals 20 and 100. Could you provide an explanation for this observation? Understanding the conditions under which OLSA surpasses the proposed method would offer insights into the limitations and potential areas for improvement.

- Could you clarify the rationale behind the factorization of the number of base predictors when evaluating Multi-layer Generalized Least Square-based Aggregation (M-GLSA)? A detailed explanation would help in comprehending the methodology and its implications on the results.

- There are typographical errors in the paper; for instance, in Line 462. It is recommended to thoroughly proofread the manuscript to correct such errors before submission. Ensuring grammatical accuracy and clarity will enhance the readability and professionalism of the paper.

---

### Note · Authors · 2024-11-15

I have read and agree with the venue's withdrawal policy on behalf of myself and my co-authors.